# Multi-Target β-Protease Inhibitors from *Andrographis paniculata*: In Silico and In Vitro Studies

**DOI:** 10.3390/plants8070231

**Published:** 2019-07-17

**Authors:** Archana N Panche, Sheela Chandra, AD Diwan

**Affiliations:** 1Department of Bio-Engineering, Birla Institute of Technology, Mesra, Ranchi 835215, India; 2MGM’s Institute of Biosciences & Technology, Mahatma Gandhi Mission, N-6, CIDCO, Aurangabad 431003, India

**Keywords:** β-amyloid, *Andrographis paniculata*, multi-target anti-amyloid agents

## Abstract

Natural products derived from plants play a vital role in the discovery of new drug candidates, and these are used for novel therapeutic drug development. *Andrographis paniculata* and *Spilanthes paniculata* are used extensively as medicinal herbs for the treatment of various ailments, and are reported to have neuroprotective properties. β-amyloid is a microscopic brain protein whose significant aggregation is detected in mild cognitive impairment and Alzheimer’s disease (AD) brains. The accumulation of β-amyloid disrupts cell communication and triggers inflammation by activating immune cells, leading to neuronal cell death and cognitive disabilities. The proteases acetylcholinesterase (AChE), butyrylcholinesterase (BChE), and beta secretase-1 (BACE-1) have been reported to be correlated with the synthesis and growth of β-amyloid plaques in the brains of AD patients. In the present study, the phenolic compounds from *A. paniculata* and *S. paniculata* that have been reported in the literature were selected for the current investigation. Furthermore, we employed molecular docking and molecular dynamics studies of the phenolic compounds with the proteins AChE, BChE, and BACE-1 in order to evaluate the binding characteristics and identify potent anti-amyloid agents against the neurodegenerative diseases such as AD. In this investigation, we predicted three compounds from *A. paniculata* with maximum binding affinities with cholinesterases and BACE-1. The computational investigations predicted that these compounds follow the rule of five. We further evaluated these molecules for in vitro inhibition activity against all the enzymes. In the in vitro investigations, 3,4-di-*o*-caffeoylquinic acid (**5281780**), apigenin (**5280443**), and 7-*o*-methylwogonin (**188316**) were found to be strong inhibitors of AChE, BChE, and BACE-1. These findings suggest that these compounds can be potent multi-target inhibitors of the proteases that might cumulatively work and inhibit the initiation and formation of β-amyloid plaques, which is a prime cause of neurotoxicity and dementia. According to our knowledge, these findings are the first report on natural compounds isolated from *A. paniculata* as multi-target potent inhibitors and anti-amyloid agents.

## 1. Introduction

β-amyloid plaques are extracellular deposits of which the major component is the β-amyloid protein (Aβ), a small polypeptide generated by processing of a much larger transmembrane β-amyloid precursor protein (APP) [1,2] through the sequential cleavage of proteolytic enzymes known as secretases [3]. The microscopic Aβ plaques found accumulated in the brain are considered the hallmark of a brain affected by Alzheimer’s disease (AD). The protease β-secretase cleaves APP at the β-site and produces soluble peptides. This is followed by another γ-secretase cleavage generating APP intracellular domains and Aβ [4]. Due to their essential roles in the generation of Aβ, both β-secretase and γ-secretase are considered to be prime targets for the development of anti-AD pharmaceuticals [5,6]. The studies on γ-secretase inhibitor drugs have faced the problem of toxicity in many clinical trials, which is thought to be related to the physiological function of γ-secretase [7]. It is may be due to a fundamental problem of lack of alternative pathways to process its natural substrates when γ-secretase is inhibited [7]. Among these two secretases, β-secretase is suggested as a more promising target, as its activity is exerted by a transmembrane aspartic protease, beta secretase-1 (BACE-1) [8,9,10,11], as well as it is a rate-limiting step in Aβ synthesis. Therefore, BACE-1 is a prominent target for anti-Aβ production. It has been reported that various BACE1 inhibitors have shown efficient reduction in brain Aβ levels [12,13,14]. Irrespective of these reports, still so far two clinical trials of mild-to-moderate AD patients have been failed due to unspecific side effects [14] as well as lack of efficacy, as per the document on the phase 2/3b EPOCH trial of Verubecestat (ClinicalTrials.gov identifier NCT01739348). Among these trials, the set of limitations of drugs included low oral bioavailability/high metabolic clearance, loss of potency in cellular systems, and inadequate access to the target localization within the central nervous system (CNS), which is often driven by P-glycoprotein or other transport systems [13]. 

AChE is an enzyme that is involved in cholinergic and non-cholinergic functions, and degrades the neurotransmitter acetylcholine (Ach), which is a helper for the transmission of messages released by the brain cells. Resembling other members of the serine hydrolase and serine protease families [15,16,17], the catalysis of AChE has been suggested as a two-step process of acylation and deacylation [18]. In early biochemical investigations, it has also been conferred that AChE induces amyloid fibril formation by interacting through its peripheral anionic site, forming highly toxic AChE-Aβ complexes [19]. It has also been observed that AChE accelerates the rate of Aβ formation, and the neurotoxicity induced by the enzyme–amyloid complex is higher than that induced by the amyloid alone [20]. Presently, the mostly used AChE inhibitors (AChEI) are physostigmine, tacrine, and donepezil [21,22,23,24], which are synthetic compounds. Galantamine is another potent AChE inhibitor that is the first compound isolated from a plant source [25,26]. However, all of these AChE inhibitors have a number of adverse effects such as hepatotoxicity and gastrointestinal complaints [27]. BChE shares 65% sequence homology with AChE and also catalyzes the hydrolysis of acetylcholine. The studies on human brains have shown the expressions of BChE in substantial populations of different parts of the brain such as the cerebral cortex, hippocampus, amygdala, and thalamic nuclei [28]. The AD progression has been found to be correlated with cholinesterase activity levels such as: if the BChE level increases, then the AChE level decreases, ultimately resulting in the decrement of acetylcholine levels, leading to the loss of cognitive function [29]. The synthetic drugs being used as inhibitors for these enzymes are associated with unavoidable side effects, such as systemic toxicity and hormonal disturbance [30,31,32,33], while natural products are known for their inherent benign effects and safety, and therefore are advantageous as therapeutics [34,35]. The AD constitutes almost 65% to 75% of all dementia cases [36]. However, at the same time, the studies also agree with the classical pathology of AD, which includes the presence of Aβ plaques and neurofibrillary tangles, accounting for 40–70% of the variation in the cognitive behaviors in elderly people, with additional pathologies such as Lewy body pathology [37] and cerebrovascular disease [38] working together with AD pathology to account for an additional 20–30% of dementia cases.

The genus Andrographis belongs to the Acanthaceae family in the major group of Angiosperms (Flowering plant), and it comprises of about 40 species. Only a few species are medicinally important, of which *Andrographis paniculata* is most popular. *A. paniculata* is commonly known as the ‘king of bitters’ or Kalmegh; it is an annual branched erect herb running 500 m to one meter in height. It is native to peninsular India, Sri Lanka, in different regions of Southeast Asia, China, America, West Indies, and Christmas Island. It has been extensively used as a medicinal herb for centuries in varieties of traditional medicine systems all over the world. It is used in Ayurveda, Unani, Siddha, as a home remedy in Indian tradition, as well as in tribal medicine in India. It is also of appreciable medicinal reputation in traditional Chinese medicine, where it is used against a variety of diseases such as diarrhea, dysentery, snakebite, malaria, fever, cold, and respiratory infections [39,40]. Experimental studies have revealed that *A. paniculata* has numerous established activities as well as anti-inflammatory and antioxidant properties that imply that it may also possess neuroprotective activities [41]. However, according to the literature survey of pharmacological studies related to the compounds from the plant against anti-Aβ target proteins, none of the reports were found to contain the desired information. 

The genus Spilanthes belongs to the family Asteraceae, which is widely distributed in the tropical and subtropical regions of the world. The plants of this genus are reported in some regions of India such as South India, Chhattisgarh, and Jharkhand. It is commonly known as toothache plant, eyeball plant, and spot plant. There are around 60 species of genus Spilanthes that have been reported from different areas and regions of the world. The genus Spilanthes is also being used as traditional medicine, and the extracts of the plants have shown antioxidant activities indicating the presence of various metabolites possessing antioxidant properties [42]. Few works of literature suggest that *S. paniculata* are proven to possess antimicrobial [43], antioxidant [44], anti-inflammatory [45], and neuroprotective activity [46]. 

Flavonoids, diterpenoids, and polyphenols are reported as the major bioactive components of *A. paniculata* [47], while alkamides have been reported as major bioactive constituents of *S. paniculata* [48]. A large number of studies have been reported targeting AChE [49,50], cholinesterases, and antioxidant activities [51,52,53] or BACE-1 [54,55] with the plant extracts. Quercetin, which is a flavonoid, has been demonstrated to possess antiamyloidogenic, BACE-1 inhibition, antioxidant activity, and amyloid fibril disaggregation ability through in vitro studies [56]. Another compound, rutin, has also been found to reduce Aβ42-induced cytotoxicity, inhibit Aβ42 fibrillization, avoid mitochondrial damage, and minimize the production of nitric oxide synthase, glutathione disulfide, and reactive oxygen species [57]. These pieces of evidence demonstrate the multi-target inhibition ability of phenolic compounds, and this information also encourages the hunt for specific compounds with well-defined pharmacological properties. Phenolic compounds have gained significant interest, as illustrated by the various reports on their efficiency in holding back a variety of human illnesses [58,59,60]. However, the complexity of the current incurable diseases has intelligibly demonstrated that single-target drug identification strategies are incompetent to achieve a desired therapeutic effect [61,62]. Simultaneously, it has also been learned that the molecules that hit more than one target might also possess by natural law a safer profile compared to single targets [61,62]. Based on this evidence, the concept of multi-target drugs has succeeded as an emerging paradigm beginning from 2000 [63,64,65], and is now one of the hot and very much needed trends in drug discovery since 2017 [66]. The scarcity of the effective drugs and the efficiency of natural compounds to exert pharmacological activity gives an efficient alternative way for the search for new multi-target therapeutic agents. 

In the present study, the compounds of *A. paniculata* and *S. paniculata* were identified from the literature, and the phenolic compounds were further investigated with molecular docking and molecular dynamics studies to explore the binding affinity with the enzymes AChE, BChE, and BACE-1, which are the potential anti-Aβ target proteins. Molecular dynamics simulations were employed to carry out the interaction analysis and validate the rationality of docking results. The compound complexes were subjected to MD simulations for 15 ns. Further, in order to validate the computational results, the in vitro enzyme inhibition assay of the compounds isolated [67] from the corresponding plant source with AChE, BChE, and BACE-1 was carried out. The current investigation is expected to obtain novel potent natural inhibitors as anti-Aβ agents that can be used as drugs for the neurodegenerative disease such as AD.

## 2. Results and Discussion

Amongst 23 metabolites, nine were found to be phenolic compounds. A total of eight phenolic compound structures could be obtained from the compound databases. The final set of phenolic compounds whose structures were obtained were selected for further studies (Figure 1). These phenolic compounds were 3,4-di-*o*-caffeoylquinic acid (CID **5281780**), apigenin (CID **5280443**), onysilin (CID **42608095**), 7-*o*-methylwogonin (CID **188316**), vanillic acid (CID **8468**), scopoletin (CID **5280460**), transferulic acid (CID **445858**), and β-sitostenone (CHEBI **68105**), which are labeled with their corresponding database IDs.

### 2.1. Molecular Docking Analysis

The phenolic compounds were all docked with proteins AChE, BChE, and BACE-1. Molecular docking discovers the efficient ligand molecule that has better binding with the target and expresses it with the H-bond binding affinity (both side chain and backbone) and Π–Π interactions. Docking is often applied to evaluate the atomic interaction association of drug candidates against protein targets to predict the binding affinity and activity of the drug. The Glide docking score is Glide’s scoring function, whose scores were used to give ranking to ligand poses obtained in docking. The scores of docking and XP docking of selected compounds with AChE, BChE, and BACE-1 are given in Table 1. The binding modes and positions of all the compounds with AChE, BChE, and BACE-1 ( Figure 2; Figure 3) were evaluated based on the nature of interactions of ligand atoms and types of residues.

It was observed that among all the compounds, **5281780** had the highest binding affinities with AChE and BChE, with global minimum energy levels of −9.770 kcal/mol and −11.946 kcal/mol, respectively (Table 1). Two poses of **5281780** showed higher binding affinities against BACE-1 with the lowest energy levels of −7.648 kcal/mol for pose 1 and −7.108 kcal/mol for pose 2, respectively. Thus, the compound **5281780** showed the highest docking scores against all the three proteins AChE, BChE, and BACE-1, implying its efficient multi-target binding potential.

The interacting residues of AChE with **5281780** were GLU292, TYR72, THR75, TYR124, PHE295, and HOH715, which formed hydrogen bonding amongst them (Figure 2). The residues TYR and THR are the aromatic, polar residues, while PHE is an aromatic hydrophobic residue. The acetylcholine binding site of AChE contains many aromatic side chains. The aromatic side chains of peripheral anionic site have been found to be crucial for fasciculins, propidium, and other specific inhibitors of AChE [68,69]. These residues, as the binding sites of **5281780**, confirm the stronger binding as well as stability of AChE and the **5281780** complex. The AChE inhibitors BW284c51, decamethonium, and E2020 (Aricept) are the known inhibitors that span both the anionic site and the peripheral anionic site of AChE, where TYR 70 (72), TYR 121 (124), and TRP 279 (286) are involved in their binding [70,71,72].

In case of BChE, **5281780** showed interactions with residues HIS438, GLY78, PRO285, and GLU197, forming hydrogen bonds and TRP82 with stronger Π–Π interactions (Figure 2). It has been well demonstrated that the differences in the activity of the same compound with AChE and BChE is due to the variations in the aromatic residues in the catalytic region of the enzymes [73]. The interacting residues are aromatic, charged, or polar, which are the major contributors in the enzyme compound interactions and binding association. These different set of residues demonstrate the variations in the activity levels of the compounds. An interesting hydrogen-bonding pattern was observed between the first pose of **5281780** and the aromatic polar residue TYR72, which formed two hydrogen bonds through a water molecule. A buried bridge was found to be formed between an important george site aromatic reside TYR and buried water molecule 715, which is the same characteristic hydrogen bonding formed through water molecule as observed in case of AChE. The hydrogen bonds are also considered as significant contributors in ligand-target binding. Furthermore, due to their directionality, they play a major role in determining the specificity of binding [74]. The hydrophobic interactions can increase the binding affinity between the ligand and the target [75]. In close observations, we can depict that the interacting residues were either polar or hydrophobic, forming hydrogen and Π–Π interactions, which signifies the strength of the binding between the ligand and the target. In case of interactions with BACE-1, the contact residues were TRP71, PHE108, GLY34, ARG128, and ARG128 for the first pose of **5281780**, and ILE126, TRP76, and TYR198 for the second pose, both forming the hydrogen bonded interactions (Figure 3). The first pose of **5281780** formed major hydrophobic interactions and it interacted with the flap containing residues ARG54-PHE108-TRP76-GLY74-PHE47 through residue PHE108. The second pose of **5281780** interacted with TYR198, which is located in the hydrophobic pocket of BACE-1. This interaction position has been seen to achieve selective BACE-1 inhibition [76].

The second top compound was **5280443**, which was also found to exhibit higher binding affinities with AChE, BChE, and BACE-1. In our present work against AChE, it showed the docking and XP scores as −7.0071 kcal/mol and −7.088 kcal/mol, respectively (Table 1). Its highest score was predicted against BChE with a docking score of −9.038 kcal/mol and XP score of −9.021 kcal/mol, respectively. It also showed higher binding affinity scores of docking as −7.422 and XP as −7.422 kcal/mol against BACE-1. The energy levels that we found in our study predict that **5280443** might be as efficient a multi-target potent candidate as that of **5281780**.

The bonding pattern of interactions of ligand atoms with these multiple proteins was evaluated; then, we found that **5280443** had major Π–Π stacking interactions rather than hydrogen bondings (Figure 1 and Figure 2). The residues TRP286 and TYR341 of AChE were observed to form three Π–Π stacking interactions with the aromatic ring of **5280443**, and two hydrogen bonds with PHE295. Tryptophans are the predominant residues in the active sites of AChE, contributing in stronger binding affinity through the Π–Π stacking. These aromatic residues act synergistically along with charged groups of ligands, showing larger incrementation in inhibition constants [77]. 

Two Π–Π stacking bonds were found to be formed: one with TRP286, which is a hydrophobic residue, while another one was formed by TYR341, which is a polar residue. It was also found that **5280443** shared two pi–pi stacking bonds with the residue TRP82 of BChE and hydrogen bonding with GLU197, THR120, and ALA328 of BChE. The residue tryptophan (TRP86 in AChE, TRP82 in BChE) is reported to play an important role in enzyme catalysis through π–cation interactions with substrates, which help align these molecules with the catalytic serine [78]. This residue is coupled with an anionic aspartate (ASP74 in AChE; ASP70 in BChE) by a polypeptide that is one of the components of a peripheral site that interacts with cationic substrates. It has been observed through studies that at high substrate levels, as the activity of AChE is decreased [79,80], the activity of BChE is increased [81]. This evidence supports the combined effect on two enzymes through a single mechanism exhibited by a compound through residue. In case of fasciculin-2, a 61-amino acid peptide from mamba venom, Van der Waals interactions with the residues TYR 20 (72), TYR 121 (124), and TYR 334 (341) are made, and it also packs in a loop through TRP 279 (286) [82]. This three-loop toxin family member has been reported to have a binding site that is highly specific for the peripheral anionic site of AChE [83]. 

In case of BACE-1 interactions, **5280443** was also found to share π–cation bonding with ARG128 and hydrogen bonding with ILE126 and TRP76 (Figure 3). These residues are polar or hydrophobic, which further strengthen the affinity predictions. The similar interactions have been reported for a compound nor-rubrofusarin 6-*O*-β-d-glucoside, which is an AChE and BACE-1 inhibitor from *Cassia obtusifolia* Linn [84].

The third top compound **188316** was also found to have strong binding affinities with the proteins AChE and BChE, and moderate affinity with BACE-1. The docking and XP scores of **188316** with AChE were predicted as −6.837 kcal/mol and −6.837 kcal/mol, while against BChE, they were −7.370 kcal/mol and −7.364 kcal/mol, respectively (Table 1). Moderate docking and XP scores of −4.872 kcal/mol and −4.879 kcal/mol against the protein BACE-1 were predicted. From the energy levels, it can be concluded that **188316** has the highest binding affinity with the cholinesterases AChE and BChE, which might lead to being a potent multi-target anticholinesterase lead candidate. The compound **188316** was also found to share a major number of Π–Π stacking interactions indicating the efficient binding. It formed Π–Π stacking interactions with residues TYR341 and TRP286 of AChE, and TRP82, HIS438, HIS438, TRP231, and PHE329 of BChE, respectively (Figure 2). Propidium, a natural AChE inhibitor, have also been reported to bind TRP286 in the aromatic peripheral anionic site [82]. It also predicted hydrogen bonding with ASN37, TRP76 of BACE-1, and shared a π–cation bonding with ARG128 and Π–Π stacking with TYR71. TYR71 is a conserved residue in the flap of BACE-1. There are reports of affinities of natural catalytic inhibitors such as nor-rubrofusarin, 6-*O*-β-d-glucoside, and 2-Amino-3-{(1R)-1-cyclohexyl-2-[(cyclohexylcarbonyl)amino]ethyl}-6-phenoxyquinazolin-3-ium with TYR71 through van der Waals interactions [84].

Many studies have highlighted the importance of hydrophobic interactions in ligand–target interactions based on the number of hydrophobic atoms present in the drugs available in the market [74]. If we compare the existence of aromatic rings among the structures **5280443** and **188316**, it can be seen that the aromatic rings exist at the same position. This kind of existence of rings at same position increases the length of the chain, the number of bond torsions, and the accessibility of hetero atoms and substitutions from the amino acid residues of the enzyme in the active site [85]. This leads to the finding that the higher number of hydrophobic interactions and hydrogen bonds existing in these complexes increases the stability of the enzyme–ligand complex. These pieces of evidence and our findings support the predictions of compounds **5281780**, **5280443**, and **188316** as the potent multi-target candidate molecules against the β-amyloid proteases. To the best of our knowledge and based on the literature survey, this is the first report of the compounds **5281780**, **5280443**, and **188316** as potent candidates that might be multi-target anti-amyloid agents from *A. paniculata*.

### 2.2. ADME Test

Lipinski’s rule of five is being traditionally used to evaluate druglikeness or the oral bioavailability of drugs in humans. It identifies five parameters: molecular mass <500 Da, lipophilicity (CLogP) <5, number of hydrogen-bond donors <5, number of hydrogen bond acceptors <10, and molecular reactivity between 40–130 [86]. Thus, the rule describes molecular properties that are crucial for ADME (absorption, distribution, metabolism, and excretion), but the rule does not predict pharmacological activity. These properties of the molecules were predicted using QikProp 3.5 [87] tool. (Table 2).

The reported qualified ranges of the available CNS and non-CNS oral drugs have been observed as: molecular weight, 75-671Da; donarHB, 0–5; acceptHB, 0–16.1; QPlogPo/w, −2.6 to 7.3; QplogBB, −3.1 to 0.78; human oral absorption, 1–3; percent human oral absorption, 10–100; rule of five, 0–2; rule of three, 0–2 [88]. According to the analysis of these qualified ranges of drug likeliness parameters, we can predict that the compound **5281780** partially satisfies the rules of five and three. The compound **5281780** with molecular weight 516.457 with slight additional weight as compared to 500 Da was observed to violate three parameters: molecular weight, donarHB, and percent human oral absorption. Due to these values, we may consider that **5281780** is marginal for the further drug development, but this should not be considered as a serious issue, and it should not be removed from further studies [89]. This has to be accepted, because the prediction tools have been designed to assess the druglikeliness ability of the compounds. There are certain compounds, for example, buserelin (MW-1299.48) and bromocriptine (MW-654.595), which can cross the blood–brain barrier, despite having a higher molecular weight [90]. At the same time, in case of identification of anti-Alzheimer’s, it has also been observed that the molecular weight criteria has been purposefully extended from <500 Da to <600Da with the purpose that the big active site of the enzyme will require a big molecule to cover and bind it appropriately [91]. With this evidence, despite violations of few drug likeliness parameters, we decided to proceed for the further in vitro investigation of compound **5281780**. The compounds other than **5281780** were found to satisfy the rules of five and three. Our two top scored compounds, **5280443** and **188316,** were predicted to have a higher percentage of human oral absorption of 74% and 100%, respectively. These two compounds were found to be with low molecular weight and better human oral absorption ability. These two parameters play crucial roles in the crossing of BBB and oral availability, ultimately increasing the drugability of the compound. With these predicted drug likeliness properties, we further validated the binding characteristics of the **5281780**, **5280443**, and **188316** compound complexes with all the three proteins AChE, BChE, and BACE-1 through MD and simulations.

### 2.3. Molecular Dynamics (MD) and Simulations

MD simulations provide every detail of conformational changes in a complex system. They help with understanding the flexibility of the model system and reveal the unseen dynamics of the complex during the interaction between the protein and the ligand molecule [92,93]. The complexes—namely, BACE1-**5281780** (pose 1), BACE1-**5281780** (pose 2), BACE1-**5280443**, BACE1-**188316**, BChE-**5281780**, BChE-**5280443**, BChE-**188316**, AChE-**5281780**, AChE-**5280443**, and AChE-**188316** were predicted to have strong binding affinities in the molecular docking studies. To further investigate and confirm the interaction profiles of these compounds with the proteins, all the complexes were subjected to MD simulations for 15 ns. To confirm the stability of the simulations, the temperature, pressure, and potential energy of the system—as well as the root mean square deviations (RMSDs) and root mean square fluctuations (RMSFs) of the C-alpha atoms of the system—were analyzed during the simulation time. The important measures of structural fluctuations in MD simulation analysis are the RMSD and RMSF [94]. The RMSD is the average displacement of the atoms at an instant of the simulation relative to a reference structure, which is usually the first frame of the simulation or the crystallographic structure. The RMSF is a measure of the displacement of a particular atom, or group of atoms, relative to the reference structure, averaged over the number of atoms. Hence, RMSF, RMSD, and H-bond interaction analysis was employed for the evaluation of the complexes.

#### 2.3.1. RMSD

The RMSD of AChE, BChE, and BACE-1 interacting with **5280443** were plotted with respect to time (Figure 4). The RMSD of AChE and BChE complexed with **5280443** showed an initial growth inclination and later showed stability after 10 ns and 5 ns, respectively (Figure 4A). The complex BACE-1 with **5280443** showed fluctuations for a quite long time, and later was stable after 11 ns. The complex AChE-**5280443** showed an average RMSD of 0.180 nm (Table 3). The other two complexes of **5280443** with BChE and BACE-1 showed an average RMSD of 0.170 nm and 0.220 nm (Table 3), respectively (Figure 4A). In case of complex AChE-**5280443**, there was an initial rise along with the fluctuations in the RMSD values, but after 3 ns, the RMSD values declined and showed stability up to 15 ns.

The complexes of **5281780** with AChE, BChE, and BACE-1 (pose 1) and BACE-1 (pose 2) showed an average RMSD of 0.191 nm, 0.176 nm, 0.175 nm, and 0.190 nm, respectively (Table 3). The complex with AChE showed a repeated pattern of rise and fall in the values followed by a stable RMSD after 12 ns (Figure 4B). The complex with BChE showed rise and fall in the RMSD values, and later it showed a stable value from 13.5 ns. The complex of BACE-1 with the first pose of **5281780** showed almost stable RMSD from 8 ns, while pose 2 showed fluctuations in the RMSD values up to 12 ns; later, it remained stable.

The compound **188316** complexed with AChE showed an initial higher RMSD followed by a fall and again rise up to 12 ns, and then after 12 ns, it retained a stable value. The other complex with BChE showed an average of 0.185 nm (Table 3) with slight deviations up to 12 ns, and later, it lowered and was stable at 0.1 (Figure 4C). On the other hand, the complex with BACE-1 had a rise and fall in its values, with an average RMSD of 0.298 nm; later, it retained stability from 9 ns. Although all the complexes showed fluctuations, these are with very minimum deviations, and almost all the RMSD values are less than 0.20 nm, except for the 0.298 nm of **188316** with BACE-1. It was also observed that all the complexes achieved a stable RMSD in the simulation run of 15 ns (Figure 4).

#### 2.3.2. RMSF

In an equilibrated model system, the structure of interest fluctuates around a stable average conformation. The computation of the fluctuations of each atom of the structure relative to the average structure of the simulation is done by RMSF [95].

The RMSFs of the C-alpha atoms of all the complexes were plotted and analyzed. The RMSFs of the C-alpha atoms for all the proteins in the complex with the compound **5280443** were found to be in the range of 0.0461 to 0.4593 nm (Figure 4D). In the complex with AChE, the highest RMSF value of 0.4593 was shown by residue at position 542, and the minimum was shown with value 0.0461.

The complex of the same compound with BChE showed comparatively higher fluctuations among all three enzyme complexes. The maximum RMSF given by this complex was 0.3 nm by the residue. The complex with BACE-1 showed fewer fluctuations than that of the other two enzymes. The highest value of fluctuation of this complex was found to be 0.32.

Upon comparing four complexes of **5281780**, it can be seen that the **5281780** complexes with BChE showed a maximum fluctuation of 5.3 nm (Figure 4E). The lowest fluctuation was shown by the compound’s second pose with BACE-1.

All the complexes with the compound **188316** shared almost the same highest fluctuation around the value of 5 nm. The maximum value of fluctuation of the complexes with AChE, BChE, and BACE-1 were observed as 5 nm, 5 nm, and 4.3 nm (Figure 4F).

#### 2.3.3. Interacting H-Bond Analysis

An analysis of the number of hydrogen bonds between the three proteins and all the phenolic compounds was carried out and were analyzed throughout the simulation period.

On average, a minimum of one and a maximum of 17 hydrogen bonds were found to be playing a role in the interaction of all the complexes. This number of intermolecular hydrogen bonds indicates stable interaction throughout the duration of MD simulations.

In the molecular docking, the protein ligand interaction analysis gives an insight on the nature of intermolecular bonding that occurred. In this analysis, it was observed that the compound **5281780** shared six, four, five, and three hydrogen bonds with AChE, BChE, and BACE-1 (pose 1) and BACE-1 (pose 2), respectively (Figure 2 and Figure 3). If we compare these predictions with the intermolecular hydrogen bondings that occurred in simulations over a span of 15 ns, it can be observed that **5281780** has shown a range of 1 to 17 hydrogen bonds (Figure 5B,E,H,I). The compound **5280443** predicted one, three, and two hydrogen bonds with AChE, BChE, and BACE-1, respectively (Figure 2 and Figure 3), whereas **188316** showed one, five, and two hydrogen bonds with AChE, BChE, and BACE-1, respectively (Figure 2 and Figure 3). The intermolecular hydrogen bonding in 15 ns span in case of **5280443** showed a number ranging from 1 to 12 (Figure 5A,D,G), whereas for **188316**, it showed a range of bonds between 0–8 (Figure 5C,F,J) for all the three target proteins. The compound **188316** showed a lower number of hydrogen-bonded interactions with the three target proteins in the docking evaluations. The same pattern of fewer bondings was also observed in the number of hydrogen bonds in the simulation results of **188316**. The compound **5281780** showed the maximum number of hydrogen bonds with all the three proteins throughout the simulation time (Figure 5B,E,H,I), while the compound **188316** showed less hydrogen bonding with all the three proteins (Figure 5C,F,J). The compound **5280443** showed a moderate amount of hydrogen bonding with all proteins (Figure 5A,D,G). The same pattern can be observed to be followed in case of the ligand–target interaction analysis done in molecular docking studies. From these findings, it can be concluded that the protein ligand complexes and their interactions were stable and stronger during the simulation time.

### 2.4. Enzyme Inhibition

Three phenolic compounds—**5281780**, **5280443**, and **188316**—isolated from *A. paniculata* were tested for the inhibition of AChE, BChE, and BACE-1. The inhibition percentage of enzyme activity against the concentration of control and isolated compounds was plotted. The inhibition constants (IC_50_) were calculated and presented as mean ± SD (Table 4).

#### 2.4.1. Cholinesterase Inhibition

The compound **5281780** extracted from the aerial parts of *Artemisia princeps* had IC_50_ values in the range of 1.78 to 2.40 μM, demonstrating a five to 10-fold greater efficacy in rat lens aldose reductase inhibition as compared to the quercetin control, which had an IC_50_ value of 17.91 μM, suggesting that this quinic acid has the potential for treating diabetic complications [96]. **5281780** and other quinic acids extracted from steamed sweet potato root have been reported to possess melanogenesis suppression ability [97,98]. Moreover, compound **5281780** extracted from the fruits of *Pandanus tectorius* had exhibited significant antihyperlipidemic activities [99]. It has also been well documented that 3,4-di-*o*-caffeoylquinic acid isolated from the aerial parts of *Lactuca indica* L. possesses the hepatoprotective property and is hence seen as a potent phytochemical agent against hepatitis B virus [100]. In our present study, **5281780** inhibited AChE and BChE with IC_50_ values of 2.14 ± 0.04 μM and 1.44 ± 0.02 μM, which were found to be lower than that of eserine with IC_50_ values of 3.39 ± 0.22 μM and 2.88 ± 0.01 μM against AChE and BChE, respectively (Figure 6 and Figure 7). According to different works of literature and comparing their IC_50_ values, it can be concluded that **5281780** in the present study exhibited better and efficient inhibition activity against both cholinesterases with lower IC_50_ values. Eserine is also called physostigmine, an alkaloid, and is one of the oldest natural AChE inhibitors, from the west African perennial shrub *Physostigma venenosum*. Another recent study showed that 3,4-di-*o*-caffeoylquinic acid, being an antioxidant, had the potential to protect human skin against environmental oxidative damage [101]. The in vitro inhibition studies of eserine on AChE (electric eel) and BChE (equine serum) exhibited IC_50_ values of 0.17 μM and 0.59 μM [102], respectively.

The second compound, **5280443**, showed inhibitory IC_50_ values of 3.42 ± 0.02 μM and 1.97 ± 0.01 μM against AChE and BChE, respectively, in comparison to eserine, which had IC_50_ values of 3.39 ± 0.22 μM and 2.88 ± 0.01 μM (Figure 6 and Figure 7). **5280443** inhibited AChE in a similar manner as that of eserine, while it also inhibited BChE with a lower IC_50_ value than that of the control compound. Thus, it can be concluded that **5280443** inhibited BChE more efficiently, and it also exhibited efficient AChE inhibition similar to eserine. The compound **5280443** has already been reported as a potent inhibitor of epidermal ornithine decarboxylase induction by 12-O-tetradecanoyl phorbol-13-acetate (TPA) in a dose-dependent manner from 1 to 20 μM in animal model studies [103]. This animal model study indicated that **5280443** not only inhibited skin papillomas, but also showed the tendency to decrease the conversion of papillomas to carcinomas. **5280443** has also been shown to inhibit antigen-specific proliferation and interferon–gamma production by murine and human autoimmune T cells in the in vitro studies [104], giving the hope for the discovery of potent natural compounds to treat autoimmune diseases. Moreover, it has been reported to have the potential to suppress the growth of pancreatic tumors with the inhibition of glycogen synthase kinase-3b with an IC_50_ of 1.9 μM [105]. The important bioactivities of apigenin isolated from red alga *Acanthophora spicifera* (Vahl) Borgesen included potent analgesic, anti-inflammatory, and anti-proliferative activities [106].

Apigenin has also been widely studied and found to be potent for anti-cancer activity for various types of cancers and having low-toxicity toward living cells because of its multiple physiological functions (antioxidant, antibacterial, anti-inflammatory, etc.) and biological effects (triggering cell apoptosis, autophagy, inducing cell cycle arrest, suppressing cell migration and invasion) [107].

The compound **188316** was found to exhibit the AChE and BChE inhibitory effects with IC_50_ values of 2.46 ± 0.03 μM and 1.46 ± 0.02 μM, respectively (Figure 6 and Figure 7). When we compared the IC_50_ values (Table 4), it was found that **188316** is more potent (almost 1.5-fold) than eserine. Amongst all the inhibitors tested, in the case of the BChE inhibition study, **188316** was shown to have twofold inhibition ability with better and efficient AChE inhibition capacity than eserine. The compound 7-*o*-methylwogonin isolated from the callus culture of *Andrographis lineata* has been proved to exhibit anticancer activity against the leukemic cell line [108]. 7-*o*-methylwogonin isolated from *A.paniculata* also exerted a dose-dependent inhibition of interleukin-1 (IL-1 ) beta production [109], providing a way to discover appropriate phytomedical treatment against a variety of inflammatory and allergic disorders.

Thus, the compounds **5281780**, **5280443**, and **188316** exhibited mean IC_50_ values of 1.44 ± 0.02 μM, 1.97 ± 0.01 μM, and 1.46 ± 0.02 μM, respectively, whereas the control showed an average IC_50_ of 2.88 ± 0.01 μM (Table 4). The synthetic derivatives of thiourea have been reported to be potent inhibitors of AChE and BChE with IC_50_ values of 8.92 ± 1.03 μM and 6.96 ± 0.961 μM, respectively [110]. Boldine, a natural alkaloid, has also been showed to possess the cholinesterase inhibition ability with IC_50_ values of 372 μmol/l and 321 μmol/l for AChE and BChE, respectively [111]. Comparing these results, it is clear that the natural compounds **5281780** and **5280443** as well as **188316** are better inhibitors of cholinesterases. The molecular docking analysis of these compounds showed stronger interactions through various important active site residues reflecting efficient binding and ultimately the inhibition ability. Thus, the computational investigations are proved to be more accurate and correlating with the experimental results. Hence, it could be concluded that the compounds **5281780**, **5280443**, and **188316** exhibited better inhibitory effects against both cholinesterases, which signifies that these compounds fairly expose their potential as dual-target anti-amyloid protein inhibitors.

#### 2.4.2. BACE-1 Inhibition

In in vitro BACE-1 inhibition studies, the compounds **5281780**, **5280443**, and **188316** showed IC_50_ values of 3.31 ± 0.12 μM, 3.79 ± 0.26 μM, and 2.91 ± 0.04 μM, respectively (Figure 8). **5281780** inhibited BACE-1 with IC_50_ values of 3.31 ± 0.12 μM, which was found to be slightly higher than quercetin, with IC_50_ values of 2.91 ± 0.03 μM (Figure 8). From these results, it can be clearly stated that **5281780** is a compatible inhibitor to quercetin. Quercetin, a flavonoid, is already known to have anti-inflammatory and antioxidant [112] properties, and has also been shown to possess neuroprotective activity in the investigations of neurodegeneration animal models [113].

The compound **188316** exhibited similar inhibitory activity as that of the control, while the other two compounds **5281780** and **5280443** exerted slightly lower activity than quercetin. Thus, we can say that these compounds have the potential to inhibit BACE-1. The BACE-1 percent values represented that **188316** inhibited BACE-1 almost in the similar manner as that of quercetin. The inhibition percentage of **188316** showed dose-dependent inhibition.

According to the results obtained, compounds **5281780**, **5280443**, and **188316** showed potent inhibition as compared to quercetin; hence, these compounds are potent multi-target inhibitors of AChE and BChE as well as BACE-1. AChE inhibitors were specifically investigated as potent therapeutic candidates for enhancing the cholinergic activities, and Indian medicinal plants are seen as the potential sources for these inhibitors [114]. These medicinal plants include *Andrographis paniculata Nees*, *Centella asiatica (L.) Urban*, *Evalvulus alsinoides L*., *Nardostachys jatamansi DC*, *Nelumbo nucifera Gaertn*, *Myristica fragrans Houtt* [114], and many more. It has been observed in the human neuronal cell line studies that the selective inhibition of BChE with cymserine analogs elevates ACh levels in the brain, reduces the secretion of the amyloid beta (Ab) peptide, and improves cognitive performance in aged rats [115,116]. The in vivo two-photon microscopy studies performed in a transgenic AD model to monitor the impact of BACE-1 inhibition on Aβ pathology showed that BACE-1 inhibition is more sensitive to Aβ initiation than that of the progression of β-amyloid pathology [117].

All these findings suggest that the inhibition of AChE, BChE, and BACE-1 will cumulatively effect the initiation and formation of β-amyloid plaques, which is a primary cause of AD. Additionally, Indian medicinal plants are considered to be the major sources for these inhibitors. Moreover, the studies performed on the effect of flavonoids on the neurotoxicity of Aβ fragments in mouse cortical cultures showed that oxidative stress was involved in Aβ-induced neuronal death, and antioxidative flavonoids strongly inhibited the Aβ-induced neuronal death [118]. Hence, flavonoids are the potent molecules that could be the strong inhibitors for Aβ pathology.

## 3. Materials and Methods

It has been quoted in Butler’s review that drugs from natural sources are equivalently effective as those of synthetic and semi-synthetic drugs, and have shown potential ability in all phases of clinical trials for almost all disease types [119]. The natural compounds from *A. paniculata* and *S. paniculata* were collected through an extensive literature survey and were segregated along with their classes. A total of 23 metabolites were found from the classes of phenolics, terpenoids, and alkamides. Polyphenols have been proven to have neuroprotective effects through various studies on cell and animal models [120,121,122,123] and have also been observed to exert neuroprotective activities for various neurodegenerative diseases [124]. Our aim was to identify the natural inhibitors that could either prevent or slow down the rate of β-amyloid production; hence, we focused only on phenolic compounds. A total of nine phenolic compounds were found from the collected dataset of compounds. Their structures were obtained from PubChem Compound [125] and the ChEBI (Chemical Entities of Biological Interest) database [126]. The compounds whose structures were not available in both the databases were unfortunately excluded. Two-dimensional (2D) structures of the compounds were sketched using ACD/ChemSketch(freeware).

### 3.1. Protein Preparation

The X-ray crystal structures of human AChE, human BChE, and human BACE-1 complexes were retrieved from the Protein Data Bank (PDB) with PDB IDs as AChE (4M0E), BChE (1POM), and BACE-1(4ZSR). These structures were prepared using Protein Preparation Wizard from Schrodinger [127,128]. The preprocessing, review of the processed structure, modifications in the structure, and final refinement of the structure were carried out to prepare each protein. The ionization states were generated; then, each protein was optimized to get the accurate structure with optimized hydrogen bonds and steric clashes. The optimization of hydroxyl, histidine, and C/N atoms was performed by flipping the position corrections and converting the HIS residue to HID or HIS using “flip”. Partial atomic charges were assigned according to the OPLS-2005 forcefield [129]. The root mean square deviation (RMSD) of all the heavy atoms was kept within “0.3°A” while doing the final minimization of the proteins.

### 3.2. Ligand Preparation

The phenolic compounds were prepared using the LigPrep [130] tool of Schrodinger. The ligand preparation was done by converting 2D structures to 3D structures, followed by the addition of hydrogen atoms, the removal of water and small ions, the generation of stereoisomers, and the generation of low-energy conformations. The Epik tool was employed to generate the protonation states of ligands at pH 7.2 ± 0.2. The ligands were optimized with the OPLS-2005 forcefield [129].

### 3.3. Molecular Docking

The rigid receptor (flexible ligand) docking of all the phenolic compounds and proteins AChE, BChE, and BACE-1 were performed using the Glide tool [131]. The Glide docking protocol of extra precision (XP) was employed while performing molecular docking of each protein with the all the known phenolic compounds. The grid box of all the three proteins—AChE, BChE, and BACE-1—were mapped covering the centroid of selected active site residues for accurate receptor and ligand interaction prediction. Receptor grids were generated, keeping the default parameters of the van der Waals scaling factor at 0.80 and the charge cutoff at 0.15 subjected. The flexible docking protocol was used for XP docking with force field OPLS_2005. For each ligand, one pose generation was allowed for one compound state. For XP flexible docking, the 10 best poses after docking were assigned for each compound. This workflow was implemented for all the three prepared proteins AChE, BChE, and BACE-1, and phenolic compounds.

### 3.4. ADME Prediction

There are very few drugs of plant origin that are being tested. For any molecule to be considered as a potent drug candidate, certain properties are needed to be possessed by it. These are absorption, desorption, metabolism, and excretion, which are known as ADME. Any molecule is tested for ADME and rule of five for its drug likeliness characteristics. The descriptors and properties of all the phenolic compounds were predicted using the QikProp [87] tool of Schrödinger. The ADME test was carried out to check the drug likeliness property of these compounds.

### 3.5. Molecular Dynamics and Simulations

To confirm the stability of the docked complexes and reveal the conformational changes in the binding mode of the ligand and the protein, molecular dynamics and simulations for 15 nanoseconds using GROMACS 4.5.1 were executed [132,133,134]. The coordinates of proteins and ligands from docking results were used for MD simulations. The protein–ligand complexes with the highest binding affinity were further tested with dynamics and simulations. The GROMOS96 43A1 force field was used for the preparation of the protein topology of AChE, BChE, and BACE-1 with pdb2gmx. The ligand topologies were generated using the PRODRG 2 server [135]. The docked coordinate positions of the ligand and protein were used for their topology generation to confirm the interaction modes. The implicit solvent system of water molecules was constructed by adding water molecules. Six Na+ ions were added to maintain the system as neutral. Later, the complex system was energy minimized using the steepest descent optimization method for 50,000 steps. The equilibration of the protein–ligand complex system was done in two phases of NVT and NPT. Each complex system was simulated for 100 ps with gradual heating from 0 to 300 K followed by energy minimization in the NVT ensemble. In the NPT ensemble, the complex system was simulated for 100 ps, and was equilibrated at 300 K with pressure coupling of Parrinello–Rahman. After the NVT and NPT equilibration, MD simulation of each protein–ligand complex was executed using the periodic boundary conditions. After the MD simulations were executed for every docked complex, the changes in protein–compound structure (RMSD), amino acid fluctuation (RMSF), and hydrogen bonds in interactions were evaluated.

### 3.6. Enzyme Inhibition Assay of Isolated Flavonoids

The compounds **5281780**, **5280443**, and **188316** with higher binding affinity and stable interactions were selected for further in vitro enzyme inhibition assay. These compounds were reported from *A. paniculata*; hence, they were isolated [67] from the corresponding plant source and further subjected to inhibition studies.

#### 3.6.1. In Vitro BACE-1 Activity Assay

The assay was performed according to the manufacturer protocol supplied in BACE-1 activity (Sigma, BACE-1FRET assay kit). The enzyme activity was observed with Varioskan Flash (version 4.00.53) using SkanIt Software (2.4.5 RE for Varioskan Flash) of NCL, Pune. The BACE-1 substrate solution (50 μM), a BACE-1 enzyme solution (2 μL), and inhibitor solutions were prepared as per the manufacturer’s protocol [6,136,137,138].

BACE-1 substrate solution was prepared with 1 mg/mL (500 μM) by adding 0.5 ml of DMSO to BACE-1 substrate. 7-Methoxycumarin-4-acetyl-Asn670, Lue 671-Amyloid β/A4 Precursor Protein 770 Fragment 667-(2,4 dinitrophenyl), and Lys-Arg-Arg amide trifluorophenyl salt was used as a substrate for BACE-1. A BACE-1 enzyme solution of ~0.3 unit/μL was prepared. The molecule quercetin was purchased from the local vendor of Sigma and was used as a known inhibitor: a control for BACE-1 inhibitors. Then, 20 μM of quercetin inhibitor solution was prepared with 1% DMSO solution. The isolated compounds were used for the isolated inhibitor solution preparation. The **5281780** solution (20 μM) with 1% DMSO was prepared after heating the solution at 55 °C for 5 min. The **188316** solution (20 μM) was prepared with 1% DMSO. The **5280443** solution was also prepared with concentration of 20 μM in 1% DMSO. These inhibitor solutions with variable concentrations (1 μM, 2 μM, 3 μM, 4 μM) were used for testing BACE-1 activities. The fluorescence assay buffer (78 μL, 79 μL, 80 μL, 81 μL), a BACE-1 enzyme solution (2 μL), the substrate solution (20 μL), and inhibitor solutions (0 μL, 1 μL, 2 μL, 3 μL, 4 μL) were added to a 96-well plate. Each concentration was considered in triplicate. The fluorescence readings were noted at 0 min as the “time zero” reading at an excitation of 490 nm and emission of 520 nm. The plate was incubated for 120 min at 37 °C, and the readings were taken. The readings were noted as S120. The percent inhibition was calculated using the formula, percent inhibition = [1 − (S_120_ − S_0_)/(C_120_ − C_0_)] × 100 [139].

#### 3.6.2. In Vitro AChE and BChE Activity Assay

The inhibitory activities of both AChE and BChE were measured using the spectrophotometric method developed by Ellman [140]. Acetylcholinesterase (Human), butyrylcholinesterase (Human) with acetylcholine iodide (ACh), and butyrylchlorine chloride (BCh) were used as their substrates, respectively. The enzyme activity was observed using Varioskan Flash (version 4.00.53), and the reading was observed using SkanIt Software (2.4.5 RE for Varioskan Flash). AChE and BChE solutions of ~0.15 unit/μL were prepared. The stock solution of AChE substrate (2.5 μM) and BChE substrate (2.5 μM) solution was prepared with DMSO, respectively. The compound eserine was purchased from Sigma vendor and was used as a known inhibitor as a control for both AChE and BChE assay. A 20-μM concentration of eserine inhibitor solution was prepared with 1% DMSO solution.

The isolated compound inhibitor solutions were prepared using the same method as discussed in the BACE-1 assay. The sodium phosphate buffer (pH 8) (78 μL, 79 μL, 80 μL, 81 μL), 20 μL of the 5, 5′-dithiobis-(2-nitrobenzoic acid) (DTNB), AChE, and BChE enzyme solution (2 μL), substrate solution (20 μL), and inhibitor solutions (0 μL, 1 μL, 2 μL, 3 μL, 4 μL) were added to a 96-well plate. Each concentration was considered in triplicate. The plate was incubated for 15 min at 25 °C, and the readings were taken at 412 nm. The percentage inhibition was calculated using the formula percentage inhibition = (1 − C/I) × 100, where C and I were the enzyme activities without and with the test sample, respectively.

## 4. Conclusions

This investigation was aimed at the identification of potent anti-amyloid inhibitors from the natural source. In the present work, the computational molecular docking and molecular dynamics of three important targets that affect the initiation and formation of β-amyloid plaques with the reported phenolic compounds of the medicinal plants *A. paniculata* and *S. paniculata* were studied. Molecular docking with SP and XP algorithms were performed in order to explore the binding possibilities and binding modes of the phenolic compounds with the potential targets AChE, BChE, and BACE-1. All the docked complexes were evaluated on the basis of the nature of bonding and the binding free energy value. The compounds **5281780**, **5280443**, and **188316** were predicted to have the highest binding affinity with AChE, BChE, and BACE-1. These compounds were found to be from the plant *A. paniculata*. All the phenolic compounds were evaluated for the drug likeliness property with QikProp tool of Schrodinger, and they were found to satisfy the recommended range of drug likeliness properties for the rules of five and three. This confirmed the drug-likeliness characteristics of the compounds **5281780**, **5280443**, and **188316**. The docked complexes of compounds **5281780**, **5280443**, and **188316** were further evaluated using MD simulations of 15 ns using GROMACS software. The RMSD, RMSF, and hydrogen bond contact results were evaluated, which reflected the stability of the complexes for a time scale of 15 ns. These docking and dynamics results confirmed the potent candidate molecules as multi-target anti-amyloid agents against AChE, BChE, and BACE.

Furthermore, in order to validate the computational predictions, these candidate molecules were tested for inhibition through in vitro enzyme inhibition studies with AChE, BChE, and BACE-1. These in vitro studies showed that the three compounds **5281780**, **5280443**, and **188316** inhibited AChE and BChE as well as BACE-1, representing themselves as the multi-target inhibitors. The three compounds **5281780**, **5280443**, and **188316** were found to be more efficient inhibitors of both cholinesterases than that of eserine, and also exhibited similar inhibitory effects against BACE-1 as that of quercetin. Thus, it can be concluded that the three phenolic compounds **5281780**, **5280443**, and **188316** from *A. paniculata* are potent multi-target inhibitors of amyloid proteins AChE, BChE, and BACE-1.

Various studies have been documented on plant *A. paniculata*, but in those studies, only andrographolide has been examined. This plant is known to have a variety of medicinal effects, but to our knowledge, there are no studies on its anti-amyloid activity nor the phenolic compounds studied in this work. Perhaps we may claim that this is the first report to study its anti-amyloid activity using the phenolic compounds as well using the computational approach for the investigation.

## Figures and Tables

**Figure 1 plants-08-00231-f001:**
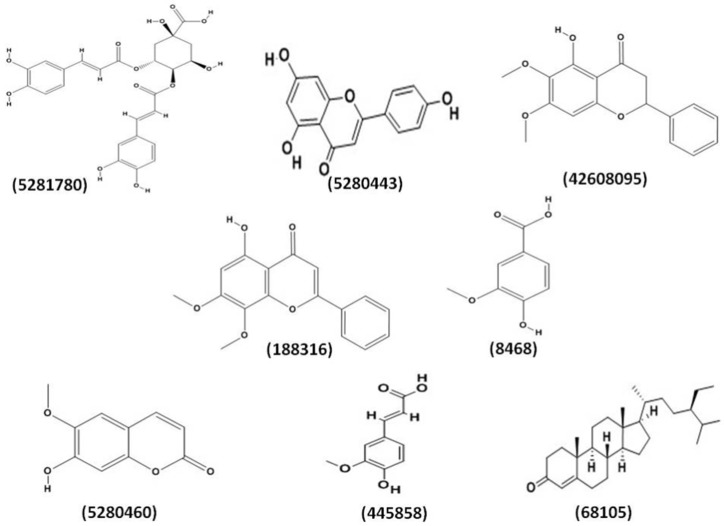
The selected phenolic compounds with their labels from *A. paniculata* and *S. paniculata*.

**Figure 2 plants-08-00231-f002:**
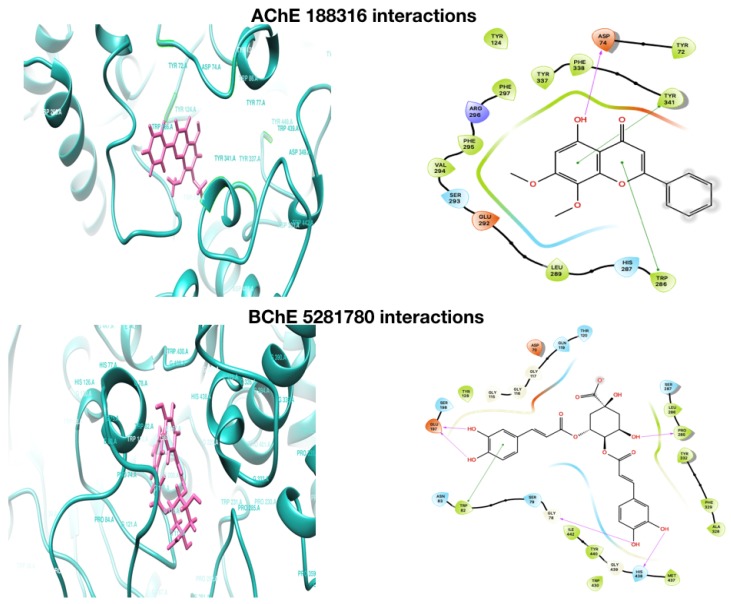
Molecular docking interaction of AChE and BChE with **5281780**, **5280443**, and **188316**.

**Figure 3 plants-08-00231-f003:**
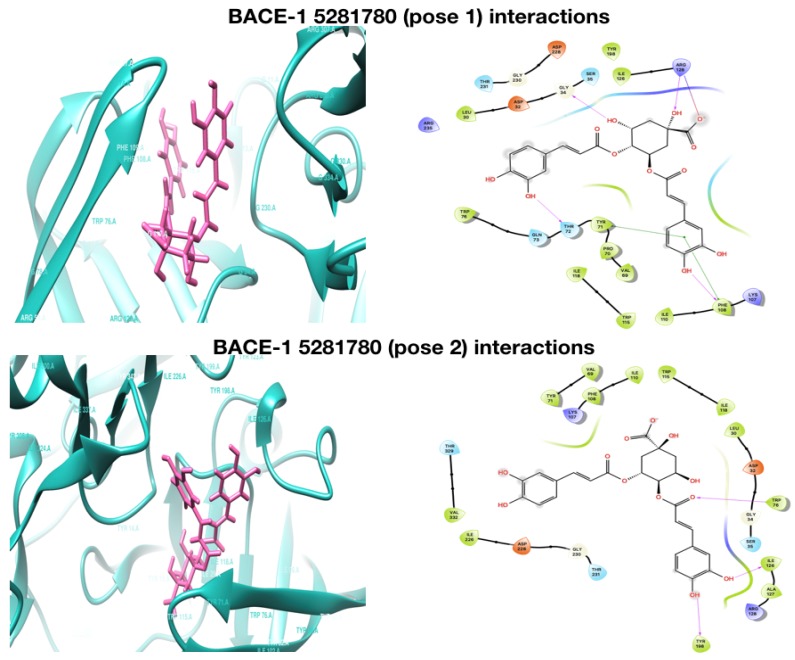
Molecular docking interaction of BACE-1 with **5281780** (pose 1, pose2), **5280443**, and **188316**.

**Figure 4 plants-08-00231-f004:**
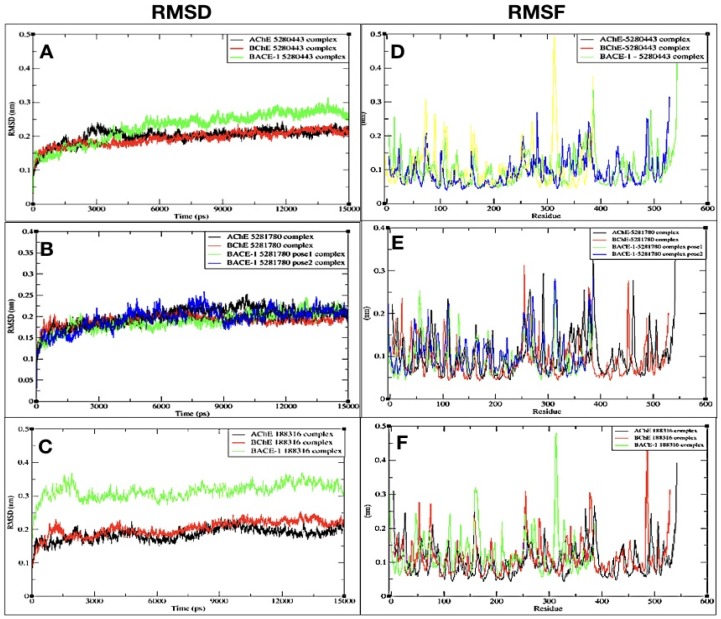
Root mean square deviations (RMSDs) of C-alpha atoms of complexes of (**A**) AChE-**5280443**, BChE-**5280443**, and BACE-1-**5280443**, (**B**) AChE-**5281780**, BChE-**5281780**, BACE-1-**5281780**-pose1, and BACE-1-**5281780**-pose2, (**C**) AChE-**188316**, BChE-**188316**, and BACE-1-**188316**, and the root mean square fluctuations (RMSFs) of residues of C-alpha atoms of complexes of (**D**) AChE-**5280443**, BChE-**5280443**, and BACE-1-**5280443**, (**E**) AChE-**5281780**, BChE-**5281780**, BACE-1-**5281780**-pose1, and BACE-1-**5281780**-pose2, and (**F**) AChE-**188316**, BChE-**188316**, and BACE-1-**188316**.

**Figure 5 plants-08-00231-f005:**
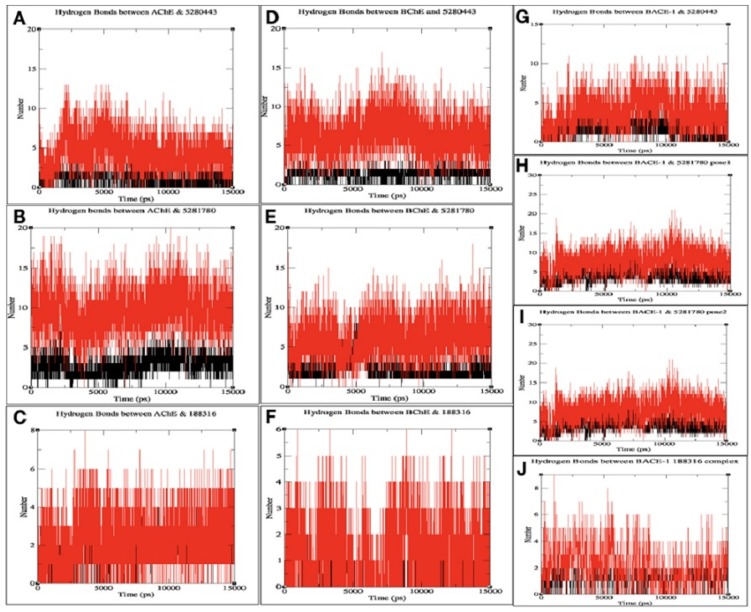
Number of H bonds between the protein–ligand complexes of (**A**) AChE and **5280443**, (**B**) AChE and **5281780**, (**C**) AChE and **188316**, (**D**) BChE and **5280443**, (**E**) BChE and **5281780**, (**F**) BChE and **188316**, (**G**) BACE-1 and **5280443**, (**H**) BACE-1 and **5281780** (pose 1) (**I**) BACE-1 and **5281780** (pose 2), and (**J**) BACE-1 and **188316**.

**Figure 6 plants-08-00231-f006:**
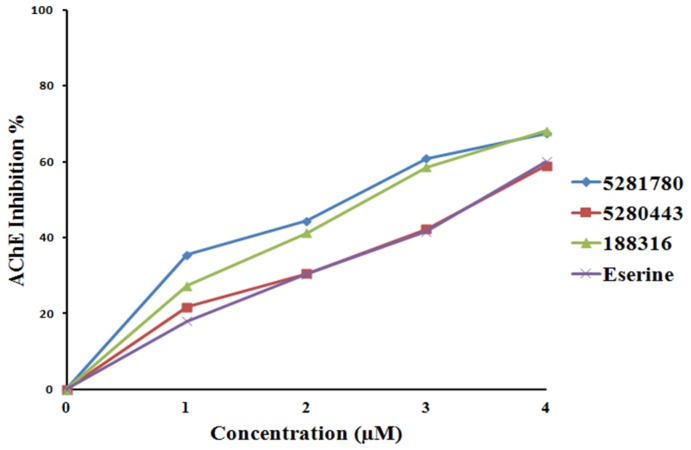
AChE inhibition percentage and inhibitor concentration plot.

**Figure 7 plants-08-00231-f007:**
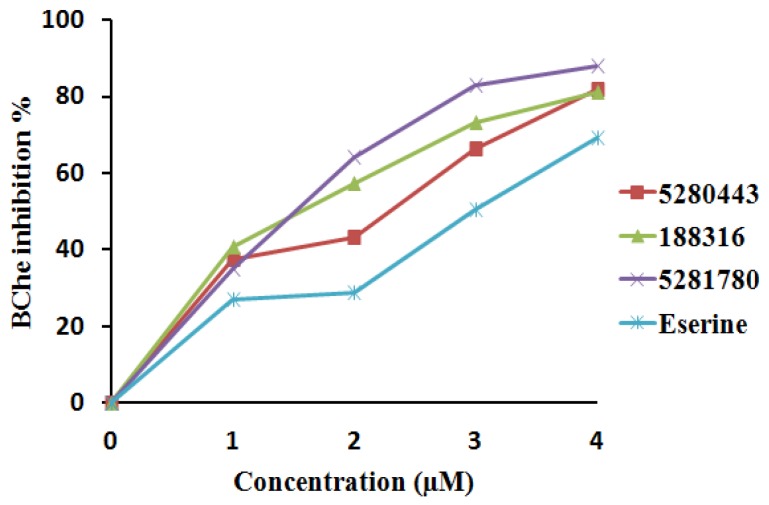
BChE inhibition percentage and inhibitor concentration plot.

**Figure 8 plants-08-00231-f008:**
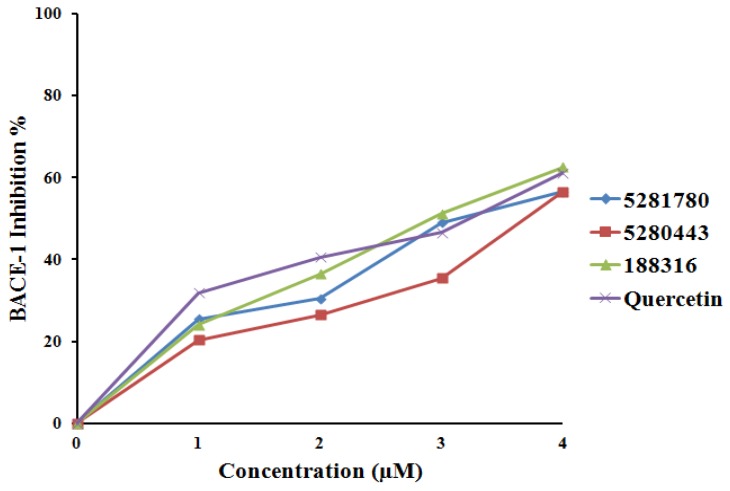
BACE-1 inhibition percentage and inhibitor concentration plot.

**Table 1 plants-08-00231-t001:** Docking score and XP score of the selected compounds with AChE, BChE, and BACE-1 obtained from Schrodinger software. AChE: acetylcholinesterase, BChE: butyrylcholinesterase, BACE-1: beta secretase-1.

Compound	AChE Docking Score (kcal/mol)	AChE XP Score (kcal/mol)	BChE Docking Score (kcal/mol)	BChE XP Score (kcal/mol)	BACE−1 Docking Score (kcal/mol)	BACE−1 XP Score (kcal/mol)
**5281780**	−9.770	−9.770	−11.946	−11.946	−7.648 (pose1) −7.108 (pose2)	−7.648 (pose1) −7.108 (pose2)
**5280443**	−7.071	−7.088	−9.038	−9.021	−7.422	−7.422
**42608095**	−6.458	−6.458	−6.915	−6.915	−4.161	−4.161
**188316**	−6.837	−6.837	−7.370	−7.364	−4.872	−4.879
**8468**	−5.303	−5.303	−5.114	−5.114	−2.966	−2.966
**5280460**	−4.905	−4.905	−5.835	−5.835	−4.424	−4.431
**445858**	−5.909	−5.909	−5.646	−5.646	−4.532	−4.532
**68105**	−7.031	−7.031	−7.082	−7.082	−3.982	−3.982

**Table 2 plants-08-00231-t002:** Selected properties predictions obtained from QikProp tool of Schrodinger for the set of compounds.

Compound	Molecular Weight	Donar HB	AcceptHB	QPlogP o/w	QPlogBB	Human Oral Absorption	Percent of Human Oral Absorption	Rule of Five	Rule of Three
**5281780**	516.457	7	11.45	0.926	−5.376	1	0	3	1
**5280443**	270.441	2	3.75	1.624	−1.411	3	74	0	0
**42608095**	300.310	0	4	3.462	−0.556	3	100	0	0
**188316**	298.295	0	3.75	3.165	−0.430	3	100	0	0
**8468**	168.149	2	3.5	1.058	−0.779	2	68	0	0
**5280460**	192.171	1	4	0.891	−0.474	3	84	0	0
**445858**	194.187	0	3.5	1.398	−1.062	3	69	0	0
**68105**	416.729	1	1.7	7.498	−0.334	1	100	1	1

**Table 3 plants-08-00231-t003:** Average RMSD values and time (ns) at which all the complexes retained stability.

Sr.No.	AChE_RMSD		BChE_RMSD		BACE-1_RMSD	
	Stable From (ns)	Average	Stable From (ns)	Average	Stable From (ns)	Average
**5281780** (pose1)	12	0.191	13.5	0.176	8	0.175
**5281780** (pose2)	-	-	-	-	12	0.190
**5280443**	10	0.180	5	0.170	11	0.220
**188316**	12	0.160	12	0.185	09	0.298

**Table 4 plants-08-00231-t004:** Mean IC_50_ and standard deviation of values of isolated compounds **5281780**, **5280443**, and **188316**, and control compounds eserine for AChE and BChE, and quercetin for BACE-1.

Sr. No.	Compound	MeanIC_50_-AChE (μM)	Mean IC_50_-BChE (μM)	Mean IC_50_-BACE-1(μM)
1	**5281780**	2.14 ± 0.04	1.44 ± 0.02	3.31 ± 0.12
2	**5280443**	3.42 ± 0.02	1.97 ± 0.01	3.79 ± 0.26
3	**188316**	2.46 ± 0.03	1.46 ± 0.02	2.91 ± 0.04
4	Eserine	3.39 ± 0.22	2.88 ± 0.01	-
5	Quercetin	-	-	2.91 ± 0.0

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
