# Peer review of "Multi-Target β-Protease Inhibitors from Andrographis paniculata: In Silico and In Vitro Studies"

_plants, 2019, doi:10.3390/plants8070231_

Round 1

Reviewer 1 Report

The manuscript entitled “Multi-target β-protease inhibitors from Andrographis paniculata: In-silico and in-vitro studies” reports on the identification of three natural compounds 3,4-di-o-caffeoylquinic acid (5281780), Apigenin (5280443) and 7-o-methylwogonin (188316) as inhibitors of the enzymes AChE, BChE, and BACE-1. Since the important role played by these proteases in the synthesis and growth of β-amyloid plaques, the identified compounds have been proposed as anti-amyloid agents against neurodegenerative diseases like Alzheimer's disease. The three natural compounds were selected from a library of phenolic molecules from A. paniculata and S. paniculata through docking studies that allowed the authors to predict their binding modes, further investigated through MD simulations. All compounds were reported to satisfy the drug likeliness properties in the ADME predictions and they displayed significant inhibition properties in in-vitroassays on the target enzymes, being sometimes more active than the reference compounds.  

The data included in the manuscript are of interest for the identification and development of drugs against neurodegenerative diseases like Alzheimer's disease. Nonetheless, there are some issues (listed below) that should be addressed before recommending the publication. Furthermore, English should be reviewed throughout the manuscript.  

Major issue:

-      The isolation procedure of the compounds used for the present investigation is not included in the manuscript (defined as unpublished results in the Introduction and Methods sections). Since the compounds were used for experimental analysis, e.g. inhibition assays on the target enzymes, the description of their isolation procedures and their characterizations should be described (otherwise the authors should refer to published procedures).  

 Minor issues:

-      Section 1. Introduction: lines 15-18, the sentence is too long (difficult to read), please simplify it (split in two shorter sentences).

-      Error in section numbering: Renumber Section 3 (and subsections) as 2 and Section 2 (and subsections) as 3.

-      Table 1 and elsewhere in the manuscript: correct “Kcal/mol” in “kcal/mol”.

-      Section 3.1 Molecular docking analysis: The section would benefit from a substantial shortening that could be supported by improved figures. The descriptions of the binding modes predicted from molecular docking analysis are poorly understandable in the current form. The author should consider to replace Table 2 and Figures 2-4 with figures showing the binding modes and the interactions predicted by their in-silicostudies. 
A discussion of the role of the residues identified as crucial for the enzyme-compound interactions could help to explain their inhibition profiles. Furthermore, a comparison with formerly reported enzyme-inhibitor complexes could highlight the importance of the interactions predicted to stabilize the compound binding.   

-      Section 3.1 Molecular docking analysis(page 8, last three lines): since the inhibition data have not been displayed and discussed yet, the assessment “potent multi-target anti-amyloid agents” is not justified only by the docking results, please modify the sentence.

-      Table 2:  the interaction with Arg128 is reported as a Π - Π stacking interaction but, at physiological pH, arginine is usually protonated. Is this a real Π - Π stacking interaction or is this a cation - Π interaction?

-      Section 3.3 Molecular dynamics (MD) and simulations: the subsection 3.3.1. RMSD would benefit from the introduction of a table the list all the values reported in the text. In this way, the extensive description of the obtained values could be avoided and the author could focus only on the most important results and their interpretation. Figure 5 and 6 are not called in the manuscript. Figure calls should be added in the appropriate positions within the section (and the subsections). In the second line of Subsection 3.3.1. RMSD, Fig. 1 is called but it should be Fig. 5 instead, please correct it.   

-      Section 3.4 Enzyme inhibition: Figures 7-9 are not called in the manuscript. Figure calls should be added in the appropriate positions within the section (and the subsections). The author should discuss the inhibition properties reported for the three compounds in light of their predicted binding modes. Is there a correlation between the predicted binding modes and the inhibition properties of the compounds? 

-      Subsection 3.4.1. Cholinesterase inhibition(page 15, lines 7-8): change “exhibited in a much better manner” in “is more potent”.

-      Section 2. Methods: which procedure is the initial paragraph describing? If this is not describing any experimental procedure it should be removed from this section.    

-      Subsection 2.1 Protein preparation: correct “0.3°A” (penultimate line).

-      Subsection 2.6.2. In-vitro AChE and BChE activity assay: “Acetylcholinesterase (Amphiphilic), Butyrylcholinesterase (Human)” (lines 2-3 of the subsection), please check the term “Amphiphilic”.  

Author Response

Dear Reviewer,

We have made changes as per your suggestions. Kindly go through it.

Thanking you.

Reviewer 2 Report

Panche, et al. describes natural product derived multi-target β-protease inhibitors from A. paniculata. It is interesting that chromone based phenolic compounds show unique activity against AchE, BChE, and BACE-1. However, I find serious issues throughout the manuscript. Please address the following concerns. 

  1. Figure 1. Remove all H atoms. It is quite disturbing the way structures have drawn.

  2. Authors have not discussed the target specificity of the molecules (No assay data or experiment results against other representative proteases).

  3. Page 5, reference 69. What is the rationale behind this reference? I don't find it is a suitable reference here. Instead, reference 69 should be located where you mentioned reference 71.

  4. Reference 71 on page 9: Authors should carefully go through the references and cite them as it is necessary. Also, I believe the following reference should be included in the manuscript.

Bioorg. Med. Chem 2012; 20(18): 5343-5351.

  5. Fig 2, Fig. 3 and Fig 4. I can't gather any useful information from those figures. Authors should provide detailed docking information (closet view of inhibitor and active site. And interacting atoms of inhibitor and amino acid side chains).  

  6.Page 9. Lipophilicity (CLogP) is a better representation than "octane/water partition coefficient".

  7. IC50 should be written as IC50.

Author Response

(The authors gave the same response as above.)

Reviewer 3 Report

Anti-amyloid inhibitors are popular therapeutic avenue for Alzheimer’s disease. In the present article, the authors have demonstrated the anti-amyloid inhibition potential of three reported compounds isolated from A. paniculata and S. paniculata. Molecular docking study have been used to demonstrate the binding ability of the three compounds with potential targets AChE, BChE and BACE-1. The drug likeliness and hydrogen bonding ability has also been demonstrated. Most of the experiments and results are well described and discussed. Any new well abundant natural product with anti-amyloid effectivity is always important for Alzheimer’s treatment. The present work will be helpful in further studies involving catechol and phenol-based anti-amyloid inhibitors. But, it will be very helpful for the readers, if some parts of the article can be more explanatory. Hence I recommend the present article to be reconsider after the following revisions:

1.       The H-Bond analysis procedure needed to be included in details. Figure 6 must be explained in details in the main text of the article. This is a very important data requires detailed discussion.

2.       The labels in the figure 5 are hard to read. Please use larger font.

3.       Figure 7, 8 and 9 are blur. Please use higher resolution pictures.

Author Response

(The authors gave the same response as above.)

Round 2

Reviewer 1 Report

The authors have made efforts in addressing the issues raised and they have improved the manuscript. I recommend the publication without requiring any further change.

Reviewer 2 Report

I believe Panche et al. is suitable for publishing in Plants journal. My only critic is in figure 1. Structures are not in the standard format.

Reviewer 3 Report

The authors have addressed all the reviewers' comments and made the necessary changes according to the suggestions made by the reviewers within the scope of the manuscript. Hence I recommend the present article to be published in its present form.